# Protocol for a prospective, observational, deep phenotyping study on adipose epigenetic and lipidomic determinants of metabolic homoeostasis in South Asian Indians: the Indian Diabetes and Metabolic Health (InDiMeT) study

Nikhil Nadiger,[1] Sarita Devi,[1] Tinku Thomas ®,[2] Ambily Sivadas,[1] Rebecca Raj-Kuriyan,[1] Sridar Govindaraj,[3] Anura V Kurpad,[1] Arpita Mukhopadhyay ®[1]

For numbered affiliations see end of article.

**Correspondence to**
Dr Arpita Mukhopadhyay;
arpitam@sjri.res.in

## ABSTRACT

**Introduction** We describe the rationale and broad study design of the Indian Diabetes and Metabolic Health (InDiMeT) study, a new prospective, observational study incorporating extensive epigenetic (DNA methylation) and lipidomic signatures to examine their association with the dysregulation of adipose de novo lipogenesis (DNL) in South Asian Indians. The InDiMeT study aims to use a case–control design to identify genetic and modifiable-environmental-lifestyle associated determinants of (1) epigenomic (DNA methylome) dysregulation of adipose DNL in type 2 diabetes mellitus (T2DM) adipose tissue, (2) identify correlates of epigenomic (DNA methylome) dysregulation of adipose DNL in peripheral blood mononuclear cells (PBMCs) from T2DM subjects and (3) elucidate plasma lipidomic correlates of adipose DNL in T2DM that can be used as biomarkers of adipose tissue dysfunction.

**Methods and analysis** The InDiMeT study will involve recruitment of 176 normoglycaemic and T2DM individuals who will be undergoing laparoscopic surgery for clinical conditions. Extensive phenotyping of the subjects will be conducted and DNA methylome and lipidomic measurements will be made. The adipose DNL pathway genes are likely to be hypermethylated in patients with T2DM with corresponding reduction of gene expression. Correlates of epigenomic (DNA methylome) dysregulation of adipose DNL pathway in PBMCs and their adipose and plasma lipidomic signatures in T2DM subjects could act as early markers of development of T2DM.

**Ethics and dissemination** For the InDiMeT study, ethical approval for addressing the specific aims has been obtained from the Institutional Ethics Committee, St John's Medical College and Hospital, St John's National Academy of Health Sciences, Bangalore. Findings from this study will be disseminated through scientific publications in peer-reviewed journals, research conferences and via presentations to stakeholders, patients, clinicians, public and policymakers through appropriate channels.

## Strengths and limitations of this study

► The InDiMeT (Indian Diabetes and Metabolic Health) study will assess the epigenome (DNA methylome)-lipidome connection in relation to their contribution to adipose de novo lipogenesis (DNL) and metabolic homoeostasis.

► The prospective case–control study design will permit us to assess the blood epigenetic (DNA methylation) and lipidomic markers that exhibit correlations with adipose DNL dysregulation and can also potentially serve as minimally invasive early biomarkers of dysglycaemia.

► Understanding the epigenetic (DNA methylation) mechanisms and their associated lipidomic profiles related to dysregulated adipose DNL in South Asian Indians will provide the rationale for effective interventional strategies targeted to this population and discriminative diagnostic/prognostic biomarkers for prevention of type 2 diabetes mellitus (T2DM) in this high-risk population.

► Though the sample size of 176 individuals will be sufficient to detect differences in adipose DNA methylation between T2DM and normoglycaemic control subjects, the primary specific aim of this study, it is unlikely to provide required sensitivity to detect dysregulation of adipose DNA methylation in specific subgroups of T2DM, such as smokers or those with chronic kidney disease.

► The subjects being recruited are the individuals who come for treatment of their pre-existing clinical conditions and could not be a random representation of the total population.

## INTRODUCTION

India is experiencing a type 2 diabetes mellitus (T2DM) epidemic, attributed to rapid sociodemographic and nutritional changes, making T2DM the 7th leading cause of death

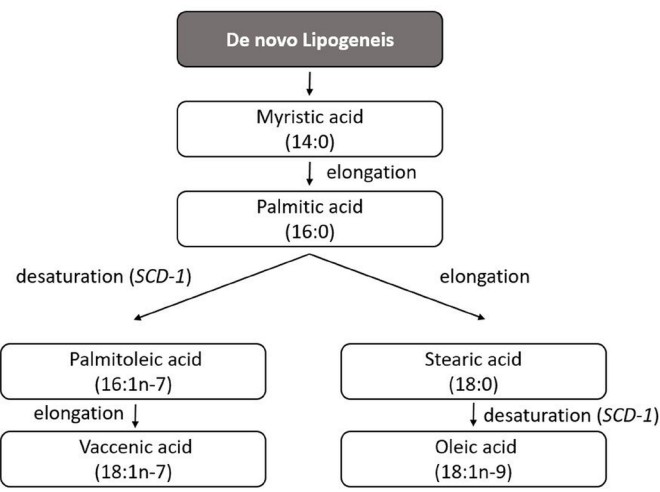

**Figure 1** The de novo lipogenesis pathway. *SCD-1*, Stearoyl-CoA Desaturase-1

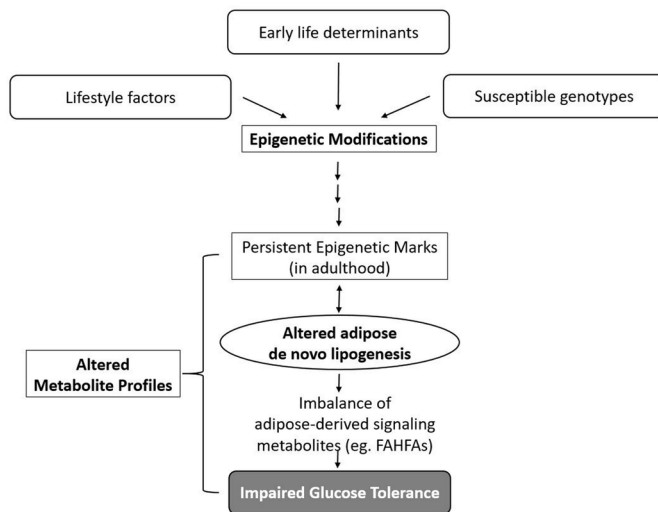

**Figure 2** The proposed link between early life determinants and lifestyle exposures on adipose DNL via epigenetic and metabolomics perturbations leading to impaired glucose tolerance. FAHFAs, Fatty acid esters of hydroxy fatty acids

overall and 4th amongst the non-communicable diseases in 2019.[1] Younger, non-obese South Asian Indians develop insulin resistance, attributed to inappropriately elevated fat accretion.[2] An important reason for elevated/ectopic fat deposition in South Asian Indians may be their uniquely high carbohydrate diets (~65 energy%), which has been associated with elevated risk of T2DM, likely due to induction of hepatic insulin resistance through increased hepatic de novo lipogenesis (DNL, figure 1).[3–5]

Unlike hepatic DNL, adipose DNL is understood to be associated with preservation of insulin sensitivity and an overall favourable metabolic phenotype ('metabotype').[6] Strawford *et al* reported negative correlations between adipose DNL and plasma glucose in humans.[7] Mice overexpressing adipose-specific GLUT4 are obese but paradoxically insulin sensitive, likely mediated by elevated adipocyte glucose influx-induced DNL.[8] Further, adipose-specific GLUT4 overexpression can rescue muscle-specific GLUT4 knockout-mediated insulin resistance, in a background of increased adiposity.[9] Therefore, it can be posited that white adipose tissue (AT) versus hepatic DNL may have opposed effects with respect to obesity-associated metabolic complications. Thus, understanding the contribution of adipose DNL in healthy versus T2DM-associated obesity is critical.

Understanding the epigenetic (DNA methylation) mechanisms and their associated lipidome profiles related to exacerbated adipose DNL in South Asian Indians will further provide the rationale for effective interventional strategies targeted to this population and discriminative diagnostic/prognostic biomarkers for prevention of T2DM in this high-risk population.

### Rationale and specific aims

The overarching goal of the Indian Diabetes and Metabolic Health (InDiMeT) study is to identify aetiologically valuable determinants that affect susceptibility to type 2 diabetes at individual, subgroup and population level by testing the hypothesis that the epigenetic (DNA

methylation) dysregulation of adipose DNL induced by high carbohydrate diet and environmental factors is an important step in the development adipose tissue dysfunction leading to type 2 diabetes (figure 2) in India. The InDiMeT study proposes to achieve this by conducting detailed phenotyping and phenome measurements in a prospective manner in a high-risk population setting. Though animal studies have underlined the epigenetic changes in the adipose associated, few have investigated adipose epigenetics in humans and none in the context of adipose DNL dysregulation and T2DM[10–12] which will help to identify epigenomic dysregulation of adipose DNL in T2DM AT. Although DNA methylation signatures are specific to tissue and cell types, only ~38% of the DNA methylation probes have been reported to be differentially methylated in blood in a tissue-specific fashion, implying likelihood of the rest ~62% probes being conservatively methylated between blood and other tissues.[13] Since blood is an easily available tissue, correlates of epigenomic (DNA methylome) dysregulation of adipose DNL in peripheral blood mononuclear cells (PBMCs) from T2DM subjects will be evaluated to validate the potential use of PBMCs as surrogates of AT dysfunction.

The overall outline of the InDiMeT study is depicted in figure 3. The specific aims of this study are:
1. To identify epigenomic (DNA methylome) dysregulation of adipose DNL in T2DM adipose tissue.
2. To identify correlates of epigenomic (DNA methylome) dysregulation of adipose DNL in PBMCs from T2DM subjects.
3. To elucidate plasma lipidomic correlates of adipose DNL in T2DM that can be used as biomarkers of adipose tissue dysfunction.

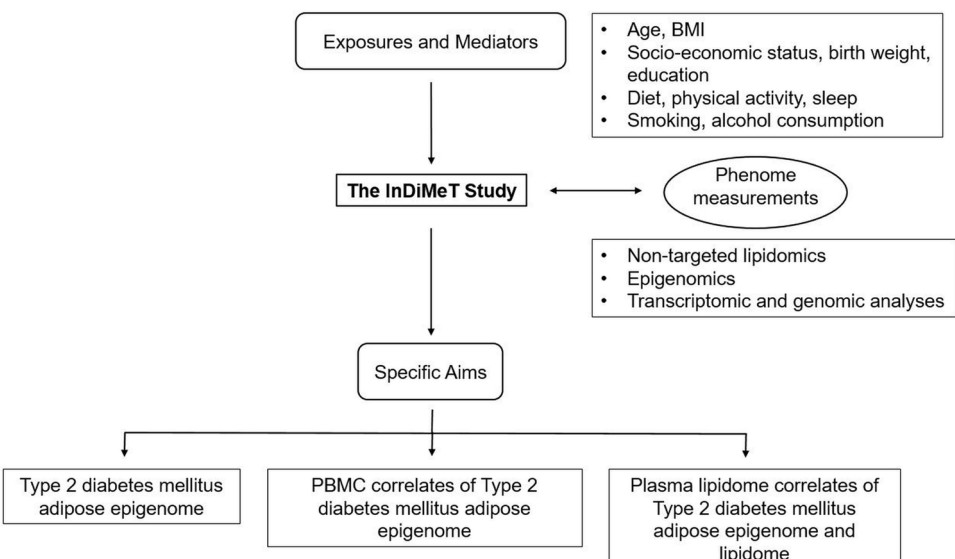

**Figure 3** The overall outline of the InDiMeT study. BMI, body mass index; InDiMeT, Indian Diabetes and Metabolic Health; PBMC, peripheral blood mononuclear cell.

## METHODS AND ANALYSIS
### Study setting
The InDiMeT study is being conducted at St John's Medical College and Hospital, a tertiary care hospital in Bangalore district, India. It is surrounded by the state of Tamil Nadu in the southeast, the Bangalore Rural district in the northeast and northwest and the Ramanagara district in the southwest.[14] As per the Indian 2011 Census, the Bangalore district recorded a decadal growth rate increase of 12% in the decade of 2001–2011 compared with the previous decade. This growth was ~4 times higher in the urban areas compared with the rural ones. The sex ratio of the Bangalore district was 916, an 8-point increase compared with the 2001 Census. The literacy rate was 88% while the work participation rate was 44%, higher for men at 62% compared with women at 25%. As per the National Family Health Survey-4, one-third of the women and 26% of the men in Bangalore (Bangalore urban district) are overweight or obese.[15] In Bangalore urban district, prevalence of elevated random blood sugar measurements stands at 8.3% among women and 10.9% among men which compares to Karnataka state-wide figures of 6.3% for women and 8.4% for men, respectively (figure 4).

### Study design and population
The InDiMeT study has been designed as a prospective tertiary care hospital-based cohort study. One hundred

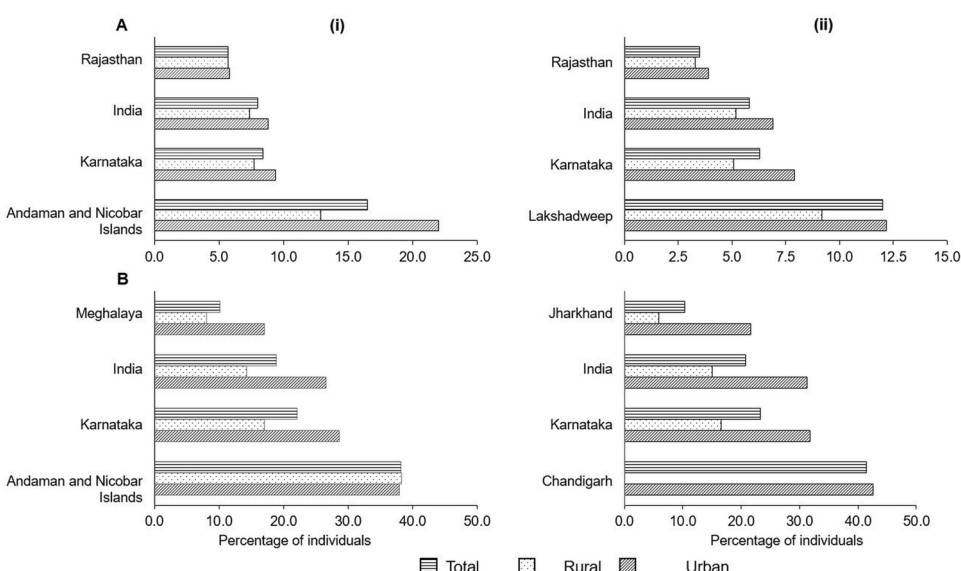

**Figure 4** Metabolic indicators. (A) Per cent of surveyed population with random blood glucose levels >140 mg/dL (measured in 15–54 year old men and 15–49 year old women) and (B) per cent of surveyed population that are overweight or obese of (i) men and (ii) women in India, Karnataka and the states with minimum and maximum values (adapted from the National Family Health Survey-4).[15]

and seventy-six normoglycaemic and T2DM individuals will be recruited into the study after explanation of details of the study and obtaining informed consent at the department of General Surgery, St John's Medical College and Hospital, St John's National Academy of Health Sciences, Bangalore. Dissemination of information about the broader study goals and likely benefits to the community as well as engagement of potential subjects from the hospital was initiated in August 2020. Recruitment of subjects commenced from 19 November 2020, after ethical approval for the study had been obtained from the Institutional Ethics Committee. Recruitment of subjects will continue until the recruitment target (176 individuals) is achieved. Strategies elaborated below are being employed for recruitment:

► Chart reviews—From the existing group of patients coming in to the Department of General Surgery, St John's Medical College and Hospital, subjects are screened and then contacted to request for their participation.

► Databases—From the database of patients coming to the St John's Medical College and Hospital, those who are eligible to participate in the study are contacted for possible recruitment.

### Inclusion criteria
Inclusion criteria are 35–60 years age; normoglycaemic and T2DM; normal and high body mass index (BMI); men and women who are undergoing laparoscopic surgeries for their clinical conditions such as cholecystectomies, hernia or splenectomy are invited to participate in the study.

### Exclusion criteria
Exclusion criteria are age outside the range of 35–60 years, individuals who are unwilling to participate in the study, individuals participating in any other study and those who tested positive for hepatitis (HBsAg: Hepatitis B surface antigen), HIV or syphilis (VDRL test: Venereal disease research laboratory test) infections. Those who have serious pre-existing medical condition, those undergoing laparoscopic surgery for clinical conditions other than cholecystectomies, hernia or splenectomy will be excluded, and these will be defined as conditions that require chronic or daily medical therapy, including medications, for T2DM, connective tissue diseases, inflammatory bowel disease, active tuberculosis and symptomatic heart disease.

### Sample size calculation
Based on the study by Nilsson *et al* the sample size required to observe a mean difference of 7% in DNA methylation in AT with an SD of 10% between T2DM and normoglycaemic controls with 90% power and 5% level of significance is 44 per group. As AT DNA methylation differences could be associated with BMI of the subjects, 44 individuals will be recruited in the normal (BMI <25 kg/m$^2$) and

high BMI (BMI ≥25 kg/m$^2$) groups of normoglycaemic controls and T2DM (35–60 years, men and women).[10]

Findings of this study could be influenced by selection bias in recruitment of the study subjects. Known sources of selection bias, such as geographical location of the study, which influences wide ranging subject parameters such as dietary diversity, levels of sunlight and air pollutant exposures, as well as recruitment of subjects from a hospital, and not from the community, are being addressed by ensuring recruitment of case and control subjects in tandem from a single study site.

### Data collection
#### Socio-demographic data collection
Detailed socio-demographic data will be collected by trained personnel using well-structured questionnaires as elaborated in box 1. The questionnaire will be provided and explained in English or local language in which the subject is most comfortable.

Socioeconomic status will be calculated based on scoring relevant for the Indian scenario.[16]

#### Medical history and medication and supplement use data collection
Participants will be asked regarding their birth weight, recent morbidity conditions and the medications and supplements they are currently taking. Details of the type and usage of these medications and supplements will be recorded by trained staff (box 1). Participants will be requested to provide these details from the prescriptions or their hospital records will be sought after.

#### Vital sign measurements
Participants will be requested to relax and then blood pressure, heart rate and pulse will be measured and recorded by trained staff.

#### Body composition measurement by DXA scanning and anthropometry
Participants will be weighed in minimal clothing using a digital scale, to a precision of 0.1 kg. The height of the subjects will be recorded to the nearest 0.1 cm. Whole body and regional body composition will also be estimated using dual-energy X-ray absorptiometry (DXA) (DPXMD 7254, Lunar Corporation, Madison, Wisconsin, USA). The total body fat (BF) will be measured and both fat and muscle (ALST mass: Appendicular Lean Soft Tissue mass) will be expressed as a percentage of body weight (% BF and % ALST).

#### Laboratory assessments
Blood samples before and after their surgery will be collected from the participants. The blood sample collected before surgery will be a fasting sample as the participants abstain from eating anything 10 hours before surgery, as a preoperative requirement for surgery. These samples will be used for assessments as elaborated in box 1. Muscle insulin sensitivity and β-cell function will be

**Box 1   Exposure and outcome measurements within the InDiMeT (Indian Diabetes and Metabolic Health) study**

**General questionnaires**

Socio-demographic

Age, ethnicity and religion, family composition, education, employment status and type, household income, socioeconomic status.

Medical history

Medical history related to birth weight, chronic and infectious diseases, last 1 month's morbidity conditions, current and past usage of tobacco (cigarettes, beedis or chewing tobacco) and alcohol, family history of diabetes and cardiovascular disease, details on most recent routine clinical investigations (haemoglobin, fasting/random/postprandial glucose, OGTT glucose, HbA1C, serum insulin and C-peptide, urea, creatinine, calcium, phosphorus, TSH, uric acid, serum cholesterol, HDL, LDL, VLDL, triglycerides, serum amylase and lipase, total protein, albumin, total bilirubin, direct and indirect bilirubin, SGOT, SGPT, alkaline phosphatase, GGT, sodium, potassium, chloride, routine urine test, urine albumin).

Medication and supplement use

Type, brand, dose and duration of medication, vitamin and mineral supplements (clinician prescribed or over the counter).

Dietary habits

3 months food-frequency questionnaire.

Physical activity

Daily, weekly and monthly frequency of sports activities, manual labour related hobbies and household chores, sedentary activities; occupational classification in terms of physical activity; sleep habits.

**Anthropometry, body composition measurements and laboratory assessments**

Vital sign measurements

Blood pressure, heart rate and pulse.

Body composition and anthropometry measurements

DXA scanning, measurement of height and weight.

Laboratory assessments

Blood investigations: Blood glucose level, HbA1C, serum insulin, C-peptide, C reactive protein, glucagon-like peptide and glucose-dependent insulinotropic polypeptide.

**Biobanking**

Collection of blood samples

Separation and storage of plasma for lipidomic analyses.

Collection of abdominal visceral adipose tissue

Done during laparoscopic surgeries; collection and storage of adipose tissue for lipidomic, epigenomic, genomic and gene expression analyses.

**Phenome measurements**

Non-targeted lipidomic analyses

On adipose tissue and plasma samples.

Epigenomic analysis

On adipose tissue samples and PBMCs.

Gene expression analysis

On adipose tissue samples.

Genomic analysis

On PBMCs.

DXA, dual-energy X-ray absorptiometry; GGT, gamma-glutamyl transferase; HbA1C, glycosylated haemoglobin; HDL, high-density lipoprotein; LDL, low-density lipoprotein; OGTT, oral glucose tolerance test; PBMC, peripheral blood mononuclear cells; SGOT, serum glutamic oxaloacetic transaminase; SGPT, serum glutamic pyruvic transaminase; TSH, thyroid-stimulating hormone; VLDL, very low-density lipoprotein.

assessed by the homoeostasis model assessment-of insulin resistance and of β-cell function, respectively.[17 18]

### Lifestyle factors

#### Dietary data collection

The dietary intake will be assessed using a validated food frequency questionnaire (FFQ). The FFQ will be administered by trained personnel to obtain information about the habitual dietary intake for the preceding 3 months.[19] The nutrients and food groups will be estimated for all the foods listed in the FFQ and summed to obtain the total nutrient or food group intake per day for an individual. Nutrient information will be assessed for 27 macronutrients and micronutrients.

#### Physical activity data collection

The habitual physical activity and sleep habits of the subjects will be assessed by detailed and validated physical activity questionnaire.[20]

### Biobanking

#### Blood collection and storage

Blood samples will be collected before and after the surgery by phlebotomy in EDTA/serum tubes, kept on ice and immediately processed for aliquoting and storage at −80°C.

#### Abdominal visceral adipose tissue collection and storage

Intra-abdominal omental adipose tissue will be collected during laparoscopic surgery with clinical indications of cholecystectomy, meshplasty (hernia) or splenectomy. The collected tissue will be appropriately stored for phenome measurements (epigenomic and non-targeted lipidomic analyses) as described below.

### Phenome measurements

#### Epigenomic (DNA methylome) analysis

Nucleic acids will be extracted from PBMCs and adipose tissue. Epigenome-wide assessment of DNA methylation levels in CpG islands will be done by reduced representation bisulfite sequencing (RRBS) after bisulfite conversion.[10 21 22]

Gene expression by quantitative real-time PCR will be performed with SYBR Green chemistry for the top 10 differentially methylated loci and for the adipose DNL pathway related genes: *MLXIPL, SREBP1, ACLY, ACC1, FASN, CPT1A, CPT1B, SCD* and *ELOVL6*. While lysis of citrate to acetyl CoA, the initial substrate of lipogenesis, is catalysed by ATP-citrate lyase (ACLY) in the cytosol, cytosolic acetyl-CoA carboxylase (ACC1) condenses acetyl CoA into malonyl CoA.[23] Fatty acid synthase (FASN) then sequentially condenses two-carbon units from malonyl CoA to form myristic acid and palmitic acid. Palmitic acid can then either undergo further elongation by elongase 6 (ELOVL6) or desaturation by stearoyl-CoA desaturate (SCD). Carnitine palmitoyltransferase 1A and 1B (CPT1A and CPT1B), SREBP1 and MLXIPL are involved in regulation of the DNL pathway. CPT1A and CPT1B are required for transport of long-chain fatty acyl-CoAs from

cytosol to mitochondria for the beta-oxidation pathway. MLX Interacting Protein Like (MLXIPL), also known as Carbohydrate-Responsive Element-Binding Protein primarily and Sterol Regulatory Element Binding Transcription Factor 1 (SREBP1), to a lesser extent, control and activate transcription of multiple DNL genes such as ACC1 and ELOVL6 in adipocytes.[24]

Genotyping of the subjects will be done using Metabochip (a custom genotyping array for metabolic and cardiovascular traits) or equivalent single nucleotide polymorphism array, such as the Infinium Global Screening Array (Illumina, USA).[25 26]

### Non-targeted lipidomic analysis

Plasma and omental AT samples will be used for non-targeted lipidomic analysis. Samples will be spiked with SPLASH Lipidomix Mass Spec Standard (Avanti Polar Lipids, USA) before lipid extraction and reconstitution. Non-targeted lipidomic analysis will be performed on a high-resolution analytical platform consisting a Vanquish Flex Binary UHPLC coupled to a Q Exactive Orbitrap, high-resolution accurate-mass spectrometer (LC-H-RAM-MS, Thermo Scientific, USA). High-resolution accurate mass data generated by Thermo Scientific Q Exactive Orbitrap will be identified and quantified by Thermo Scientific Lipid Search software. The following lipid species are of special interest: palmitic acid, stearic acid, oleic acid, myristic acid, palmitoleic acid, acyl carnitines, palmitic acid esters of hydroxyl stearic acids, furan fatty acid metabolite 3-carboxy-4-methyl-5-propyl-2-furan propanoic acid.[27–30]

### Assessment of adipose DNL

Assessment of adipose DNL will be done by determining AT triglyceride (TG) content of the fatty acids myristic acid (14:0), palmitic acid (16:0) and stearic acid (18:0) that have been reported to be correlated with plasma non-esterified fatty acids (NEFA) 14:0 and 18:0 but not with plasma NEFA 15:0, that is exclusively derived from dietary dairy sources.[31] Further, AT lipid species (from non-targeted lipidomic data) that positively correlate with transcript abundances of adipose DNL pathway genes will be identified as indicators of adipose DNL. It should be noted here that though adipose tissue myristic acid, palmitic acid and stearic acid contents are not direct measures of adipose DNL, they represent end products of DNL and have been reported to be correlated to expression of genes involved in adipose DNL.[31]

### Plasma NEFA correlates of adipose DNL

Plasma NEFA correlates of adipose DNL will be identified in the T2DM and normoglycaemic subjects by correlation analysis of plasma lipidomic data with adipose DNL measures (AT TG content of the fatty acids myristic acid (14:0), palmitic acid (16:0) and stearic acid (18:0)). This is a novel strategy for estimating adipose DNL and identifying plasma correlates of adipose DNL as such indirect methods do not exist at present and the direct

stable isotope based method of measuring DNL primarily measures postprandial hepatic DNL rates.[32]

### Data analysis plan

Continuous data will be presented as either mean±SD or median (quartile 1 and quartile 3). The normality of distribution will be examined using the Shapiro-Wilk test and Q-Q plots. Demographic and clinical characteristics will be compared between cases of T2DM and controls by Student's t-test, if normally distributed or Mann-Whitney U test, if non-normally distributed, for continuous variables and by $X^2$ test for categorical variables.

For analysis of RRBS data, the read sequences from the RRBS libraries will be adapted/quality trimmed using Trim Galore[33] and aligned to the reference human genome using a wildcard mapper like BSMap.[34] DNA methylation scores (β values) will be calculated from the aligned reads.[35] Differentially methylated position (DMP) analysis will be conducted using limma software while adjusting for cell proportions (for PBMCs) and including age and sex as covariates.[36] Statistical significance will be set at $p<0.05$ following adjustment for multiple testing using the Benjamini-Hochberg false discovery rate (FDR).[37] Differentially methylated regions will be defined as three or more contiguous probes within a 2 kb distance, each sharing the same direction of methylation change, each achieving a Benjamini-Hochberg FDR corrected $p<0.05$ in DMP analysis. Enrichment analysis will be performed to understand the top significantly deregulated gene ontologies and pathways. Additionally, in-depth analysis of methylation status of the CpG islands in the promoter and coding regions of genes associated with the DNL pathway will be conducted.

Association studies between the CpG methylated regions and clinical traits will be performed using linear mixed-model package pyLMM and significant associations will be determined using Bonferroni corrected p value cut-offs.

Gene expression values will be estimated by normalising target gene expression against expression of house-keeping genes followed by comparison of target gene expression between the groups.

Methylation quantitative trait loci will be identified by testing for association between methylation levels and genotypes of the samples using FDR correction.

Correlation-based clustering analysis of the lipid profiles will be performed and visualised as heatmaps to evaluate the inter-relationships between the lipid species and for identifying signatures that differentiate between T2DM and control samples using supervised and unsupervised approaches. For heatmap analysis, only significantly altered features (significant alterations will be defined) will be analysed and data will be organised using a Euclidean distance map and complete clustering algorithm. Random forest will be used to assess the classification performance of the lipidomics signatures. The significantly perturbed pathways will be identified based on the differentially regulated lipids. For random forest

and pathway impact analyses, data will be filtered using the IQR function and scaled using auto-scaling, in which values are mean-centred and divided by the SD of each variable. For the random forest analysis, the number of predictors to try for each node will be set to the square root of the total number of variables.

To identify the adipose lipid species that correlate with transcript abundances of genes in the adipose DNL pathway, first correlation analysis (Pearson's correlation, for normally distributed data and Spearman's correlation, for non-normally distributed data) will be conducted. This will be followed by regression analysis, with levels of lipid species exhibiting significant correlations as dependent variable and transcript abundance of individual genes in the adipose DNL pathway as independent variables, adjusting for probable confounders such as BMI and sex, if found to be significantly correlated with the dependent variables.

All statistical analysis will be processed and analysed using a combination of SPSS V.25 (IBM), Stata V.14 (StataCorp) and R (R Foundation for Statistical Computing, Vienna).

## Patient and public involvement

Patients and general public were not invited to comment on the study design and were not consulted to formulate study outcomes. Results for the InDiMeT study will be communicated to the patients and public in two complementary ways. Data related to individual participants of the study will be shared with them, if deemed beneficial for improving their health status, in consultation with clinician collaborators on this study and specific participant or their primary care providers. This is mandated by our Institutional Ethics Committee as well. The overall study results will be disseminated to participating individuals and public via appropriate forums for public science outreach.

## ETHICS AND DISSEMINATION
### Ethical considerations and informed consent

Institutional ethical approval for the study has been obtained from the Institutional Ethics Committee (IEC), St John's Medical College and Hospital, St John's National Academy of Health Sciences, Bangalore (IEC Study Ref No. 246/2020, IEC Study title: 'De novo lipogenesis in adiposity and glycaemic control: Epigenetic mechanisms and associated metabolite profiles'). Subject recruitment has been initiated from 19 November 2020.

The objectives and aims of the study are being explained in subject's preferred local language to interested volunteers who fulfil the inclusion criteria. The patient information sheet contains relevant and specific information about the study, collection and storage of samples, genetic analysis and safe storage of subject data. A written, informed consent is being obtained from the subjects before the start of the protocol.

## Data storage, management and protection

Data and resources generated as part of this study will be managed, stored and shared in line with institutional, state and national policies. In brief, specifics are as follows:

### Organisation

Conventions for naming and storage organisation of data files as well as samples from human subjects (adipose and blood) will be set up, documented and circulated among study team members at the beginning of the study.

### Storage

Data will be stored in a secure manner in three copies in two different media. Back-up copies of data will be accompanied with metadata that describe the stored data in a standard, understandable format. Data storage and protection support will be obtained from the Institutional Information Technology team.

### Physical samples

Samples from the study subjects will be processed and stored as per the experimental protocols. Samples in excess of the requirements of the study objectives will be stored appropriately as per the institutional policies as well as policies of the peer-reviewed journals where findings from the study will be submitted for possible publications.

## Dissemination of data

Sequence data generated from this study will be deposited in GenBank and epigenetic data in Gene Expression Omnibus, while raw and processed lipidomic data will be deposited in appropriate repositories such as Metabolomics Workbench or MetaboLights during the time of publication. Findings from the InDiMeT study will be appropriately communicated in refereed journals and forums for scientific and community science outreach.

## SIGNIFICANCE AND OUTLOOK

The InDiMeT study will provide the base and act as a robust tool for mechanistic interpretations of public health and epidemiological aspects of developing T2DM and its complications in the Indian setting. Though a few large, prospective cohort studies to identify the determinants of T2DM are being conducted in India, to the best of our knowledge, none of them involve stated deep phenotyping or extensive phenome measurements of the participants.[38 39] This study was conceptualised to fill this niche in this country. The carefully monitored and quality-controlled collection and archiving of biological samples, along with availability of anthropometric, socio-demographic, nutritional and medical data on the participants will aid toward planning and implementation of an array of hypothesis-driven mechanistic studies.

The InDiMeT study will not be without limitations. Notable ones will include the low probability of participation by individuals of specific subsets of the population due to the voluntary nature of participation. The

subjects being recruited are the individuals who come for treatment of their pre-existing clinical conditions could not be a random representation of the total population. Further, the study setting being a tertiary care hospital in a large, urban area in India, a country known for its large regional variations in cuisine and culture-associated dietary habits, composition and amounts of habitual dietary intakes of the study subjects are likely to be heterogenous. These factors could act as confounders and limit our ability to detect differences in DNA methylation and lipid profiles that are of low effect sizes. Detailed recording of these factors as proposed, will however help us in understanding if these factors do act as confounders in our study and allow us and others to improve design of similar studies in future.

**Author affiliations**
[1]Division of Nutrition, St John's National Academy of Health Sciences, Bangalore, Karnataka, India
[2]Department of Biostatistics, St. John's Medical College and Hospital, Bangalore, Karnataka, India
[3]Department of General Surgery, St John's Medical College and Hospital, Bangalore, Karnataka, India

**Acknowledgements** The contribution of research fellow Bajanai Nongkhlaw, research assistant Beena Bose and nutritionist J Jayakumar is gratefully acknowledged.

**Contributors** The authors of the manuscript and their individual contributions towards the manuscript are as follows. AM, TT and AVK: designed the research; NN, SD and SG: conduct of the study; AM, TT, AS, RR-K and AVK: analysis of the data, performing statistical analyses and interpretation of the results; AM and AVK: obtained funding; AM: had the idea for and led writing of the manuscript and had primary responsibility for the final content of the manuscript; and all authors critically reviewed the manuscript, read and approved the final version of the manuscript.

**Funding** This study is supported by the Wellcome Trust/DBT India Alliance Fellowship awarded to AM (grant number IA/CPHI/19/1/504593) and to AVK (grant number IA/M/14/1/501681) and by the Department of Biotechnology, Government of India (Grant Sanction order BT/PR24342/PFN/20/1314/2017 to AM and AVK).

**Competing interests** None declared.

**Patient and public involvement** Patients and/or the public were not involved in the design, or conduct, or reporting, or dissemination plans of this research.

**Patient consent for publication** Not required.

**Provenance and peer review** Not commissioned; externally peer reviewed.

**ORCID iDs**
Tinku Thomas http://orcid.org/0000-0002-1786-6076
Arpita Mukhopadhyay http://orcid.org/0000-0001-8260-5385

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
