## [Reviewer comments · BMJ Open]

ARTICLE DETAILS

TITLE (PROVISIONAL)	Protocol for a prospective, observational, deep phenotyping study on adipose epigenetic and lipidomic determinants of metabolic homeostasis in South Asian Indians: The Indian Diabetes and Metabolic Health (InDiMeT) Study
AUTHORS	Nadiger, Nikhil; Devi, Sarita; Thomas, Tinku; Sivadas, Ambily; Raj-Kuriyan, Rebecca; Govindaraj, Sridar; Kurpad, Anura; mukhopadhyay, arpita

VERSION 1 – REVIEW

REVIEWER	Lizcano, Fernando Universidad de La Sabana, Biosciences
REVIEW RETURNED	21-Oct-2020

GENERAL COMMENTS	the study project: Protocol for a prospective, observational, deep phenotyping study on adipose epigenetic and lipidomic determinants of metabolic homeostasis in South Asian Indians: The Indian Diabetes and Metabolic Health (InDiMeT) Study. - It is an ambitious study that has a clinical and Molecular Biology approach.- It seeks to detect biological markers that put at risk of suffering from Diabetes Mellitus 2 in a specific population in southern India. The study is well structured, it is ambitious in its objectives as mentioned previously and it seems to me that it is a very interesting research challenge for this population that has an increasing increase in type 2 diabetes mellitus every day. There are some conceptual aspects that would be worth correcting. In the terms used to describe the scope of work, epigenetic factors are always referred to. It should be noted that epigenetics encompasses a broad concept that includes covalent modifications of Histones and non-coding RNAs. For this reason, I consider that the term DNA methylation should always be used in this study, which is appropriate for the objectives set out in this work.- I believe that the protocol should better describe the function of the genes chosen to be studied for the adipose DNL pathway.- It would be interesting to have the lifestyle data of the patients entering the study. Given that there may be heterogeneity in the population, it is possible that the difference in DNA methylation patterns is not so evident because the number of patients participating in the study is small.
---

REVIEWER	Xiang, Angie Monash University, Public Health and Preventive Medicine
-----------------	--

REVIEW RETURNED	26-Nov-2020
-------------

GENERAL COMMENTS	The proposed study aims to describe epigenomic and lipidomic signatures of adipose tissue and venous blood in normoglycaemic versus T2DM individuals. The major concern in the proposed study design is the lack of recognition for confounding factors affecting the above, compounded by a relatively small sample size for an omic-based study. The inclusion criteria is broad - encompassing a wide age range, both sexes and all BMIs, and there is no stratification of patients with common comorbidities such as dyslipidaemia, chronic renal disease or smoking - all of which would affect lipid metabolism. Furthermore, although participants will maintain a food questionnaire, it seems an oversight that the proposed study does not include any period of diet standardisation given the well recognised effects of dietary intake on lipid flux. Discussion regarding the choice of non-targeted lipidomic analysis over targeted lipidomics would also be of interest. Particularly given the physiological significance of low abundance acylcarnitines in de novo lipogenesis, it is surprising that the authors would forgo targeted analysis which would most accurately quantify such lipids. Lastly further discussion regarding the methodology for quantifying adipose de novo lipogenesis and its plasma products would be appreciated (page 16, lines 10-36). It would appear that the authors are using a non-validated means of performing the above, and further justification would be prudent.
---

VERSION 1 – AUTHOR RESPONSE

Reviewer: 1

Dr. Fernando Lizcano, Universidad de La Sabana

Comments to the Author: the study project:

Protocol for a prospective, observational, deep phenotyping study on adipose epigenetic and lipidomic determinants of metabolic homeostasis in South Asian Indians: The Indian Diabetes and Metabolic Health (InDiMeT) Study.

- It is an ambitious study that has a clinical and Molecular Biology approach.
- It seeks to detect biological markers that put at risk of suffering from Diabetes Mellitus 2 in a specific population in southern India.

The study is well structured, it is ambitious in its objectives as mentioned previously and it seems to me that it is a very interesting research challenge for this population that has an increasing increase in type 2 diabetes mellitus every day.

Response: We thank Dr. Lizcano for recognizing the importance of conducting this study in the Indian population.

There are some conceptual aspects that would be worth correcting. In the terms used to describe the scope of work, epigenetic factors are always referred to. It should be noted that epigenetics encompasses a broad concept that includes covalent modifications of Histones and non-coding

RNAs. For this reason, I consider that the term DNA methylation should always be used in this study, which is appropriate for the objectives set out in this work.

Response: We agree with Dr. Lizcano on the need of specifying DNA methylation as the representative epigenetic phenomenon that we will be evaluating in our study. We have made the necessary changes in the manuscript to reflect this.

- I believe that the protocol should better describe the function of the genes chosen to be studied for the adipose DNL pathway.

Response: We had not elaborated on the genes related to the adipose DNL pathway and their functions as we are planning to conduct epigenome-wide assessment of DNA methylation levels in CpG islands by reduced representation bisulfite sequencing (RRBS) after bisulfite conversion. Based on Dr. Lizcano's suggestion, we have now included description of function of these genes [Page 13, lines 226-237].

- It would be interesting to have the lifestyle data of the patients entering the study. Given that there may be heterogeneity in the population, it is possible that the difference in DNA methylation patterns is not so evident because the number of patients participating in the study is small.

Response: Lifestyle data of the subjects, including physical activity, occupational classification and sleep habits, will be recorded through questionnaires as detailed in Table 1. We agree with Dr. Lizcano's opinion that owing to heterogeneity within type 2 diabetic subjects and even normoglycemic subjects, we might not be able to detect DNA methylation differences of smaller effect sizes and we have included this as a limitation of our proposed study [Page 20, lines 386-390].

Reviewer: 2

Dr. Angie Xiang, Monash University

Comments to the Author:

The proposed study aims to describe epigenomic and lipidomic signatures of adipose tissue and venous blood in normoglycaemic versus T2DM individuals.

The major concern in the proposed study design is the lack of recognition for confounding factors affecting the above, compounded by a relatively small sample size for an omic-based study. The inclusion criteria is broad - encompassing a wide age range, both sexes and all BMIs, and there is no stratification of patients with common comorbidities such as dyslipidaemia, chronic renal disease or smoking - all of which would affect lipid metabolism.

Response: We agree with Dr. Xiang regarding the potential existence of multiple confounding factors in our study. We have addressed two of the known confounding factors, namely, sex and BMI of the subject, by aiming to recruit both men and women subjects and by recruiting subjects in a stratified and statistically powered manner: normal and high BMI categories [Page 8, lines 146-151]. We are also recording detailed information on lifestyle factors of the subjects, including physical activity, occupational classification, sleep habits and their medical history. These details will eventually help us identify the factors that act as confounders for our analysis on DNA methylation and lipidomics.

As far as sample size of our study is concerned, we have included relatively small sample size as a limitation of our study. We had used data from an earlier study by Nilsson et al to calculate sample size for our study.[1] Nilsson et al had reported array-based epigenome-wide DNA methylation analysis in adipose tissue from two type 2 diabetic and normoglycemic case-control cohorts with 50 cases and 70 controls and 28 each of cases and control subjects respectively. Similarly, other studies in the recent past have reported findings from epigenome-wide DNA methylation analysis on human adipose tissue with sample sizes similar or lower to our study. For instance, Rönn et al compared array-based epigenome-wide DNA methylation in adipose tissue from 15 men with family history and 16 men without family history of T2DM while Huang et al compared adipose tissue and blood array-

based epigenome-wide DNA methylation in 143 individuals.[2,3] As such, even though likely existence of confounding factors and the relatively small sample size might not let us identify omics-based differences between T2DM and controls that are of small effect sizes, we should be able to identify the differences with large effect sizes. This will provide us and others with a handle to understand in future mechanistic studies the cellular and inter-organ causal pathways and networks that get dysregulated en route to developing T2DM and in epidemiological studies and intervention trials the lifestyle modifications that can counter such dysregulation from setting in.

Furthermore, although participants will maintain a food questionnaire, it seems an oversight that the proposed study does not include any period of diet standardisation given the well-recognised effects of dietary intake on lipid flux.

Response: We have intentionally not included any period of diet standardisation as the subjects will be recruited for this study from the Department of General Surgery, St. John's Medical College and Hospital from amongst individuals who will be undergoing laparoscopic surgery for their clinical conditions, for whom diet standardisation prior to the surgery will not be feasible. However, considering all such individuals abstain from eating anything 10 hours before surgery, we are unlikely to see acute effects of variations in diet on lipid flux [Page 12, lines 187-188].

Discussion regarding the choice of non-targeted lipidomic analysis over targeted lipidomics would also be of interest. Particularly given the physiological significance of low abundance acylcarnitines in de novo lipogenesis, it is surprising that the authors would forgo targeted analysis which would most accurately quantify such lipids.

Response: Though we have chosen to conduct non-targeted lipidomic analysis in order to characterize adipose tissue and plasma lipid profiles of T2DM subjects at a global level, we are aware of the pitfalls of such an approach, including comparatively lower likelihood of detecting low abundance species such as acylcarnitines. Since acylcarnitines are physiologically relevant species for the DNL pathway as correctly pointed out by Dr. Xiang, we have included them in the study protocol as lipid species of special interest and we will take a targeted approach for them and any of the other lipid species of interest, if we are unable to detect them appropriately in our non-targeted analysis [Page 14, lines 250-253].

Lastly further discussion regarding the methodology for quantifying adipose de novo lipogenesis and its plasma products would be appreciated (page 16, lines 10-36). It would appear that the authors are using a non-validated means of performing the above, and further justification would be prudent.

Response: Available methods of directly measuring DNL in blood using stable isotopes primarily measure hepatic DNL. As such, we plan to evaluate plasma lipidomic correlates of adipose DNL measures (adipose tissue triglyceride contents of myristic acid, palmitic acid steric acid). Though adipose tissue myristic acid, palmitic acid and steric acid contents are not direct measures of adipose DNL, they represent end products of DNL and have been reported to be correlated to expression of genes involved in adipose DNL.[4] We have elaborated on this in the manuscript accordingly [Page 14, lines 254-256; page 15, lines 257-262].

References

- 1 Nilsson E, Jansson PA, Perfilyev A, et al. Altered DNA Methylation and Differential Expression of Genes Influencing Metabolism and Inflammation in Adipose Tissue From Subjects With Type 2 Diabetes. *Diabetes* 2014;63:2962–76. doi:10.2337/db13-1459
- 2 Huang Y-T, Chu S, Loucks EB, et al. Epigenome-wide profiling of DNA methylation in paired samples of adipose tissue and blood. *Epigenetics* 2016;11:227–36. doi:10.1080/15592294.2016.1146853
- 3 Rönn T, Volkov P, Davegårdh C, et al. A Six Months Exercise Intervention Influences the Genome-wide DNA Methylation Pattern in Human Adipose Tissue. *PLoS Genet* 2013;9:e1003572. doi:10.1371/journal.pgen.1003572

4 Roberts R, Hodson L, Dennis AL, et al. Markers of de novo lipogenesis in adipose tissue: associations with small adipocytes and insulin sensitivity in humans. *Diabetologia* 2009;52:882–90. doi:10.1007/s00125-009-1300-4